# Aluminium(III) Oxide—The Silent Killer of Bacteria

**DOI:** 10.3390/molecules28010401

**Published:** 2023-01-03

**Authors:** Mateusz Schabikowski, Paweł Kowalczyk, Agnieszka Karczmarska, Barbara Gawdzik, Aleksandra Wypych, Karol Kramkowski, Karol Wrzosek, Łukasz Laskowski

**Affiliations:** 1Institute of Nuclear Physics Polish Academy of Sciences, 31-342 Kraków, Poland; 2Department of Animal Nutrition, The Kielanowski Institute of Animal Physiology and Nutrition, Polish Academy of Sciences, 05-110 Jabłonna, Poland; 3Institute of Chemistry, Jan Kochanowski University, 25-406 Kielce, Poland; 4Centre for Modern Interdisciplinary Technologies, Nicolaus Copernicus University in Toruń, 87-100 Toruń, Poland; 5Department of Physical Chemistry, Medical University of Bialystok, 15-089 Białystok, Poland; 6Department of Heart Diseases, The Medical Center of Postgraduate Education, 01-813 Warszawa, Poland

**Keywords:** anodic aluminium oxide, Fpg glycosylase, oxidative stress, bacterial *E. coli* strains, antibiotics

## Abstract

In this article, we describe the antimicrobial properties of pristine anodised aluminium oxide matrices—the material many consider biologically inert. During a typical anodisation process, chromium and chlorine compounds are used for electropolishing and the removal of the first-step aluminium oxide. Matrices without the use of those harmful compounds were also fabricated and tested for comparison. The antibacterial tests were conducted on four strains of *Escherichia coli*: K12, R2, R3 and R4. The properties of the matrices were also compared to the three types of antibiotics: ciprofloxacin, bleomycin and cloxacillin using the Minimal Inhibitory Concentration (MIC) and Minimum Bactericidal Concentration (MBC) tests. Moreover, DNA was isolated from the analysed bacteria which was additionally digested with formamidopyrimidine-DNA glycosylase (Fpg) protein from the group of repair glycosases. These enzymes are markers of modified oxidised bases in nucleic acids produced during oxidative stress in cells. Preliminary cellular studies, MIC and MBC tests and digestion with Fpg protein after modification of bacterial DNA suggest that these compounds may have greater potential as antibacterial agents than the aforementioned antibiotics. The described composites are highly specific for the analysed model *Escherichia coli* strains and may be used in the future as new substitutes for commonly used antibiotics in clinical and nosocomial infections in the progressing pandemic era. The results show much stronger antibacterial properties of the functionalised membranes on the action of bacterial membranes in comparison to the antibiotics in the Fpg digestion experiment. This is most likely due to the strong induction of oxidative stress in the cell through the breakdown of the analysed bacterial DNA.

## 1. Introduction

Anodised aluminium oxide (AAO) has been well known for many years [1,2]. It is a self-organised, highly porous material with regularly ordered pores and, consequently, a high specific surface area [3,4]. Self-organised nanoporous materials, in general, have great potential to be used in a broad range of applications [5] with functionalisation [2,6,7] or additional coatings [8] giving rise to new possibilities. One of the impressive applications of such nanocomposites is the new-generation biocidal materials which may be applied as disinfecting surfaces in both public spaces and medical facilities [9,10,11,12]. Such action in times of the SARS-CoV-2 pandemic and hospital-acquired infections seems to be highly justified [13,14,15,16]. The biocidal properties of such materials were described in numerous publications; however, to our surprise, the antibacterial properties of pristine AAO seem to be not well discovered. Typically, the antibacterial properties of aluminium(III) oxide are reported for its nanoparticle form [17]. In contrast, some publications describe AAO matrices as neutral to microorganisms [10,18]. This statement, however, contradicts the results obtained in our experimental work: The pristine AAO matrices are definitely not inert to bacteria. For this reason, studying the antimicrobial properties of pure anodised alumina oxide seems to be a vital task.

Moreover, anodised aluminum oxide matrices can also be used in systems where the anticoagulant, hemodynamic and redox properties of water are analysed—soluble derivatives of Piloty’s acid which can efficiently release azanone (HNO) at pH 7.4 and 37 ${}^{\circ }$C (HNO donors). Thrombotic events are one of the leading causes of morbidity and mortality worldwide [19,20]. Therefore, intensive research is needed to develop new agents for the prevention of thrombotic events. Based on model bacterial systems, anodised aluminum oxide may be helpful in understanding the mechanisms regulating thrombotic events and in searching for new drugs capable of modulating molecular interactions between the endothelium and blood cells. They can also participate in gas exchange as new potential gas transmitters in the modulation of physiological processes related to hemostasis, hemodynamics and even metastasis [21].

In this consistent study, we undertake this topic and analyse the antimicrobial action of anodised aluminium(III) oxide membranes in detail. To be more thorough, we studied the response of four strains of *Escherichia coli* bacteria. Moreover, the tests were conducted for three antibiotics, ciprofloxacin (cipro), bleomycin (bleo) and cloxacillin (clox) and compared to the AAO matrices.

The experimental material undergone within the framework of this study was AAO matrices containing approximately 34–47 nm pores, arranged in regular 2D hexagonal order as shown in Figure 1. In order to exclude the influence of any harmful reagents on the antimicrobial properties of AAO, we also investigated two reference materials. They were similar AAO matrices with omitted one or two steps of the synthesis in order to avoid using chlorine or chromium which could potentially influence the results even in a trace amount. Such simplified syntheses resulted in a slightly different structure of materials. For details, see Section 2.

## 2. Materials and Methods

The anodised aluminium(III) oxide membranes were prepared in three ways:A standard two-step anodisation with the use of perchloric acid (HClO${}_{4}$, 60% pure p. a., Chempur, Poland) and chromium (VI) oxide (CrO${}_{3}$, pure p. a., Chempur, Poland) compounds for electropolishing and the removal of the first-step Al${}_{2}$O${}_{3}$, respectively. This sample is denoted as *AAO Cl+Cr*, and its structure can be seen in Figure 1.A matrix prepared in a one-step anodisation procedure with the omission of the preliminary anodisation to avoid the necessity of the removal of aluminium oxide layer with the use of the chromium compound. This sample is denoted as *AAO Cl*,A matrix prepared without electropolishing in a one-step anodisation procdure. The fabrication of this type of sample requires none of the chlorine and chromium compounds. This sample is denoted as *AAO Clean*.

The procedure for the fabrication of *AAO Cl+Cr* can be found in our previous work on the antibacterial properties of AAO membranes with chemically attached functional groups [11]. In short, aluminium foil (99%, SmartMembranes, Halle, Germany) was cut into 3 cm × 3 cm squares and heated at 550 ${}^{\circ }$C for 12 h in a nitrogen atmosphere. Next, the foils were *electropolished* in a perchloric acid solution (HClO${}_{4}$, 60 % pure p. a., Chempur, Piekary Śląskie, Poland) at 20 V for 15 min.

The *preliminary anodisation* was conducted in 0.3 M oxalic acid (C${}_{2}$H${}_{2}$O${}_{4}$× 2H${}_{2}$O, a standard solution, prepared from 99.5 % pure reagent, (Chempur, Poland) at 20 V for two hours. This process is done to create a regular template on the surface. Next, the anodised aluminium oxide layer was removed in a CrO${}_{3}$ solution at 60 ${}^{\circ }$C for two hours.

The main anodisation procedure was performed according to the same procedure as the *preliminary anodisation* with the exception that its duration was extended to 24 h.

The *AAO Cl* sample was prepared with the omission of the *electropolishing* step to avoid using the perchloric acid solution.

Finally, the *AAO Clean* sample was prepared without *electropolishing* and *preliminary anodisation* steps. The washed aluminium foil was used after the thermal treatment only for the main anodisation at the same conditions as previous samples.

The membranes were analysed regarding their morphology and chemical composition using scanning electron microcopy (SEM) TESCAN VEGA3 (TESCAN, Czech Republic) and energy-dispersive X-ray spectroscopy (EDS). The micrographs were taken at 7 kV and 12 beam intensity at the distance from the sample of 4 mm. EDS measurements were performed at 10 kV, the magnification of 200× and the distance from the sample of 12 mm. The diameters and pore distance were calculated by manually measuring at least 100 of each from SEM micrographs using ImageJ v. 1.53t [22].

### 2.1. Microorganisms and Media

*Escherichia coli* strains R1–R4 and K-12 were obtained as a kind gift from Professor Jolanta Łukasiewicz at the Ludwik Hirszfeld Institute of Immunology and Experimental Therapy (Polish Academy of Sciences). The reference bacterial strains of *E. coli* (K12 ATCC 25404, R2 ATCC 39544, R3 ATCC 11775, R4 ATCC 39543) were provided by LGC Standards (U.K.) and were used according to the recommendation of ISO 11133: 2014. These strains were used to test the antibacterial activity of the synthesised agents [20,21,22,23,24,25,26,27,28,29,30]. Bacteria were cultivated in a tryptic soy broth (TSB; Sigma-Aldrich, Saint Louis, MI, USA) liquid medium and on agar plates containing TSB medium at 25 ${}^{\circ }$C. Alternatively, TSB agar plates were used. The specific growth rate (μ), according to first-order kinetics, was measured using a microplate reader (Thermo, Multiskan FC, Vantaa, Finland) at 605 nm in the TSB medium. Lanes 1kb-ladder and Quick Extend DNA ladder (New England Biolabs, Ipswich, MA, USA) were used for MIC and MBC tests as described in detail in our previous works [20,21,22,23,24,25,26,27,28,29,30] and analysed by the Tukey test indicated by * *p* < 0.05, ** *p* < 0.1, *** *p* < 0.01. Model strains of *E. coli* were plotted in all 48-well plates observed; K12, R2-R4 which were treated with the analysed compounds. From the analysis of the MIC and MBC assays, colour changes were observed for all the compounds tested but at different levels and at different dilutions. Bacterial strains R3 and R4 were the most susceptible to modification with these compounds due to the increasing length of their LPS (visible dilutions of 10${}^{-3}$ corresponding to the concentration of 0.25 μM) compared to K12 and R2 strains (visible dilutions of 10${}^{-6}$ corresponding to a concentration of 0.06 μM). The analysed R4 strain was the most sensitive, most likely due to the longest length of the lipopolysaccharide chain in the bacterial membrane. In all cases, the MBC values were approximately 170 times higher than the MIC values.

### 2.2. Estimation of Minimum Inhibition Concentration and Minimum Bactericidal Concentration

A drug or compound (visible dilutions of 10${}^{-2}$ corresponding to the concentration of 0.0015 μM) directly kills the vegetative forms of the bacteria. The obtained results depict that at all studied concentrations, AAO have an inhibitory effect on each studied bacterial model. Varied inhibitory activity was noted depending on the nature of the substituent in the aromatic ring of the tested compounds.

MICs and MBCs were determined using the methodology described previously [20,21,22,23,24,25,26,27,28,29,30] with serial dilutions of 10${}^{-1}$ to 10${}^{-7}$. In short, the analysed model *E. coli* strains were grown in a TSB medium and diluted in water. Next, 50 μL TSB medium was inoculated with 10${}^{6}$ cfu/mL (CFU, colony-forming units, approximately 0.5 McFarland units) of the bacterial strains. Next, the bacteria were transferred onto AAO matrices and incubated with each at a concentration of 1 mM for 24 h at 37 ${}^{\circ }$C. Next, 50 μL of the bacterial suspension from all the analysed membranes was diluted to 0.1 mM and plated on a 48-well plate (according to the markings) with resazurin (0.02 mg/mL), which was added to all the wells, as an indicator. The plates were incubated at 30 ${}^{\circ }$C for 24 h. Colour changes from blue to pink or yellowish with turbidity were considered positive. The lowest concentration at which there was no visible colour change was set as the MIC.

The MBC was estimated based on the measurement of the bacterial activity after a 24 h incubation with or without the AAO matrices. The lowest concentration at which there was no visible red colour was taken as the MBC.

An example of the analysis of test compounds of various concentrations using a resazurin dye on microplates (mg/L) is presented in Appendix A. Experiments were performed with three independent replicates.

The bacterial DNA was isolated from cultures of K12, R2, R3 and R4 *E. coli* as described in [20,21,22,23,24,25,26,27,28,29,30].

All microorganisms and media were precisely described in the previous works [20,21,22,23,24,25,26,27,28,29,30] and analysed by the Tukey test indicated by (*p* < 0.05): * *p* < 0.05, ** *p* < 0.1, *** *p* < 0.01, Table 1). ANOVA tests were used for the statistical analysis of the biological assay results using the JMP software [31]. The error bars in the graphs represent the standard deviation of the appropriate result.

## 3. Results and Discussion

### 3.1. Microscopic Observations

The anodised membranes were characterised just after the syntheses using SEM regarding their morphology (Figure 2). The anodisation without electrochemical preparation of a substrate resulted in the formation of irregular pores covering only parts of the surface. Stripes present on the surface deriving from the foil processing influence the location of the pores due to different surface energy levels (Figure 2a). The average size of the pores and the distance between them are the smallest from other membranes with the largest deviation (relative to the size, Table 2). Electropolishing aluminium foil prior to anodisation results in significantly more pores covering the whole surface (Figure 2b). The one-step anodisation produced relatively irregular pores (in comparison to the two-step anodisation): Twin or triplet pore formation in one cell occurs more often. The increased average distance between pores compared to *AAO Clean* indicates the formation of proper anodisation cells for each pore or pore system (twins, triplets). Finally, the membranes fabricated with the use of the two-step anodisation method consist of fairly ordered pores within each aluminium grain. The size of the pores is almost the same as for the electropolished membrane. The average distance between the pores is the same as for the one-step anodisation membrane. However, the standard deviation is approximately 54% smaller indicating significantly more orderly placed pores on the surface.

The EDS measurements revealed no chlorine and no chromium contamination in any of the samples. The minimal values for the two elements visible in Table 3 are the result of forcing the software to search for their corresponding peaks. The identified gold derives from the evaporated conductive layer for SEM studies. The values are presented as detected without rounding for visibility.

### 3.2. Antibacterial Properties

The model bacterial strains of *E. coli* K12, R2-R4, were treated with the analysed anodised aluminum oxide using MIC and MBC assays with 48-well plates. Colour changes were observed for all tested compounds but at different levels and at different dilutions. The most susceptible to modification with these compounds were the bacterial strains R3 and R4 due to the increased length of their lipopolysaccharides (LPS, visible dilutions 10${}^{-2}$ corresponding to the concentration of 0.0015 μM) in comparison to strains K12 and R2 (visible dilutions 10${}^{-6}$ corresponding to the concentration of 0.0015 μM and 0.02 μM for K12 and R2, respectively, Figure 3 and Table 1). The analysed R4 strain was the most sensitive of all strains most likely due to the longest length of the lipopolysaccharide chain in the bacterial membrane which may curl relaxing depending on the used material.

The analysed K12 and R2–R4 strains of *Escherichia coli* are not only the dominant species of the human aerobic bacterial microbiota and microbiota of various habitats in which people live, e.g., bathrooms, clinics and hospitals, but they can also temporarily colonise the oropharynx and skin of healthy people. However, apart from saprophytic strains (harmless to humans), there are also strains of *E. coli* which are pathogenic for humans and cause various forms of acute diarrhoea. Infection usually occurs through contaminated food and water, as in the case with other bacterial diarrhoea, and (less frequently) through indirect contact. In industrialised countries, pathogenic enteric strains of *Escherichia coli* are rarely part of the intestinal microbiota of healthy people; therefore, they are considered strictly pathogenic. When the appropriate amount of bacteria is ingested by a person susceptible to pathogenic infection, strains of *E. coli* have the ability to cause inflammation of the small and/or large intestine. Adequate gastric acidity has a disinfecting effect and protects (to a certain extent) against infection. Therefore, people with low gastric acidity are particularly susceptible to infections with pathogenic *Escherichia coli* strains. The source of the infection is a sick person or a vector (except for STEC/EHEC strains, for which cattle are the source). Intestinal diseases caused by pathogenic strains of *E. coli* occur in the form of epidemics or, in rare cases, with an increase in the incidence in the summer months which is the rule for bacterial diarrhoea. Therefore, we studied these strains with the use of the AAO matrices to investigate their etiology and the mechanism causing their resistance to many known and commonly used antibiotics. Despite the number of reports in the literature on the pharmacological and biological properties of AAO matrices, their antibacterial activities are still being rediscovered. Thus, there exists a need for additional research on their cytotoxic effect on selected hospital bacterial strains causing diseases associated with blood infections such as sepsis. The AAO matrices still need to be further studied *in vivo* to better delineate the pharmacological potential of this class of substances, so there is a need to clarify their role [23,24,25,26,27,28,29,30,32,33,34].

In all the analysed cases, the MBC values were approximately 260 times higher than the MIC values (Figure 3). Based on the observed results in the MIC and MBC tests, it was found that the analysed AAO matrices significantly influenced the fragmentation of the membrane and the structure of the cell wall of bacteria containing LPS of various lengths [23,24,25,26]. This causes high oxidative stress in the cell which affects the damage and modification of bacterial DNA caused by the analysed anodised aluminum oxide compounds. This was additionally confirmed by the digestion with the specific enzyme Fpg of modified bacterial DNA (Labjot, New England Biolabs, UK) which is a marker of oxidative stress [27,28,29,30,32,33,34,35] recognising oxidised guanine and adenine (Figure 4).

The highest values after the application of the membranes in model strains of *E. coli* after the use of MIC, MBC, and MBC/MIC tests were observed after treatment with a mixture of chlorine with chromium and with chlorine alone. They were increased in all strains and comparable with each other after the use of chlorine alone or chlorine with chromium and their highest value was observed for the R4 strain whose LPS length was the longest and was relaxed inducing significant oxidative stress in the cell. A similar effect was observed after the strains were modified with the AAO membranes and treated with the Fpg protein.

In reference to our previous work related to the analysis of functionalised anodised aluminum oxide [11], we observed a clear effect of their action on selected bacterial strains. On the other hand, compared to the analysed nonfunctionalised anodised aluminum oxide, we found similar but lower values for the model bacterial slippers. This may be due to the fact that the LPS, contained in the bacterial membrane, under the influence of the functionalised membranes can fold in a specific way creating conformations that are pulled out of the cell on the “flip out” principle. This drastically and rapidly changes the electrokinetic potential of the membrane increasing the level of stress oxidative activity in the cell which disrupts its homeostasis.

Located in the bacterial membrane, the outward LPS covers part of it by the “passivation” principle, which can greatly facilitate the accessibility of a given compound to interact with and act on it (Figure 5) because the R4 strain has the longest LPS and the most labile, as described in detail in the work of Prosts [26] and Maciejewska [25]. A similar effect was observed in the works of Kowalczyk [23] and Borkowski [24] after treatment with quaternary ammonium ionic liquids on selected model strains of *E. coli* where a clear change in the electrokinetic potential of the membrane was observed. The interaction of the applied functionalised membranes on the surface of the *E. coli* cell already generates strong oxidative stress in it, where there is a strong fragmentation of bacterial DNA after treatment with the Fpg protein from the group of repair glycosylases, as it was described later in the results and their discussion.

The obtained values of oxidative damage after digestion with Fpg protein in bacterial DNA were compared with modifications of bacterial DNA after treatment with antibiotics: ciprofloxacin, bleomycin and cloxacillin [23,24,25,26]. The presented research shows that the analysed compounds will be used in the future as typical “substitutes” for new drugs in relation to antibiotics used in hospital infections. The obtained MIC values, as well as our previous studies with various types of the analysed compounds [23,24,25,26], indicate that they show a strong cytotoxic effect on the analysed model bacterial strains K12 and R2–R4 (Figure 6).

Based on the MIC and MBC values, the analysed compounds were selected for further studies (on the basis of their highest biological activity similar to antibiotics) for the analysis of oxidative stress in the cell by modifying their bacterial DNA. Using the Fpg protein, we wanted to observe whether the resulting modifications in bacterial DNA would introduce oxidative damage to the DNA strand by altering the topological three forms of bacterial DNA; ccc, oc and linear forms. As observed in previous studies [27,28,29,30,32,33,34,35], the results of bacterial DNA modified with the analysed compounds (Figure 5) showed that all analysed compounds with different types of localised substituents can strongly change the bacterial DNA topology, even after digestion with Fpg protein based on agarose gels.

Changes in the main topological forms of the plasmid: ccc, oc and linear were observed in DNA isolated from model strains and digested with Fpg protein. Approximately 3.5% of oxidative damage was identified after digestion with the Fpg protein which indicates that the analysed compounds damage bacterial DNA very strongly similar to the observations of previous studies [27,28,29,30,32,33,34,35] in the tested bacterial strains, in particular R4, as evidenced by the obtained MIC and MBC values which were statistically significant at *p* < 0.05 (Figure 6 and Figure 7).

## 4. Conclusions

Anodic aluminium oxide membranes exhibit strong antibacterial properties comparable to some of the tested antibiotics. Despite no detected chlorine and chromium content, the standard two-step anodisation membrane, *AAO Cl+Cr*, is the most efficient material. The toxicity of the analysed membranes with the appropriate strains of bacteria depends on their interaction with the membrane, which may affect the destruction of cell walls by changing their hydrophobicity. Changes in the permeability and integrity of the bacterial membrane may result in a specific bacterial response to biologically active compounds, e.g., antibiotics used. This may suggest that the presence of these compounds has a toxic effect on bacterial LPS, generating strong oxidative stress, which we observed in our previous studies [20,21,22,23,24,25,26,27,28,29,30]. The results of the presented studies are important for understanding the biological properties of the tested membranes as a function of potential new antibiotics such as kanamycin, streptomycin, ciprofloxacin, bleomycin and cloxacillin, commonly used in nosocomial infections with pathogenic strains causing various diseases: syphilis, sepsis or psoriasis. The use of membranes will multiply their toxic effect on Gram-negative and Gram-positive bacteria in the face of the growing drug resistance pandemic.

Based on our knowledge, we can precisely design special membranes with high selectivity of tissue functions. Currently, one of the most important research tasks is a broader understanding of the mechanisms of interaction of these compounds on various structures, both at the cellular and molecular level. Particular attention should be paid to their use in air conditioners located in offices, hospitals or food rooms.

There is still a need for a more fundamental understanding of the performance of already synthesised and newly designed membranes *in vivo* using high-throughput bacterial screening in both Gram-positive and negative systems. Another practical application of the different membranes will be understanding their physiological effects under environmental conditions. In order to introduce them commercially to the market, further research is required on their biostability, half-life, preferred method of administration and potency on the analysed bacterial, fungal or viral strains.

## Figures and Tables

**Figure 1 molecules-28-00401-f001:**
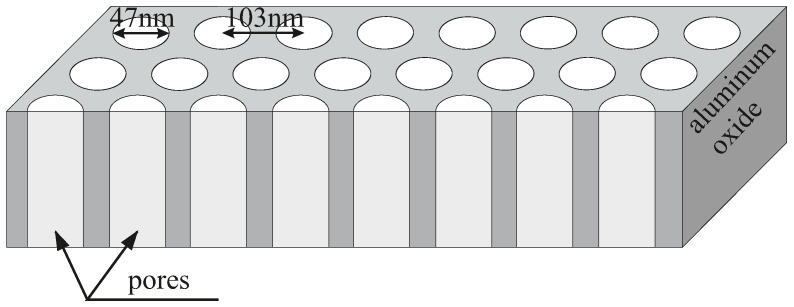
The schematic representation of the structure of an anodised aluminium oxide membrane.

**Figure 2 molecules-28-00401-f002:**
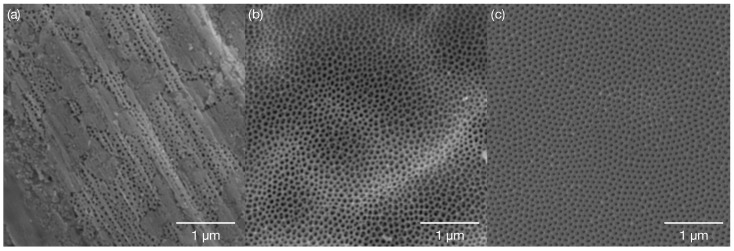
Scanning electron micrographs of the anodised membranes: (**a**) *AAO Clean*, (**b**) *AAO Cl* and (**c**) *AAO Cl+Cr*.

**Figure 3 molecules-28-00401-f003:**
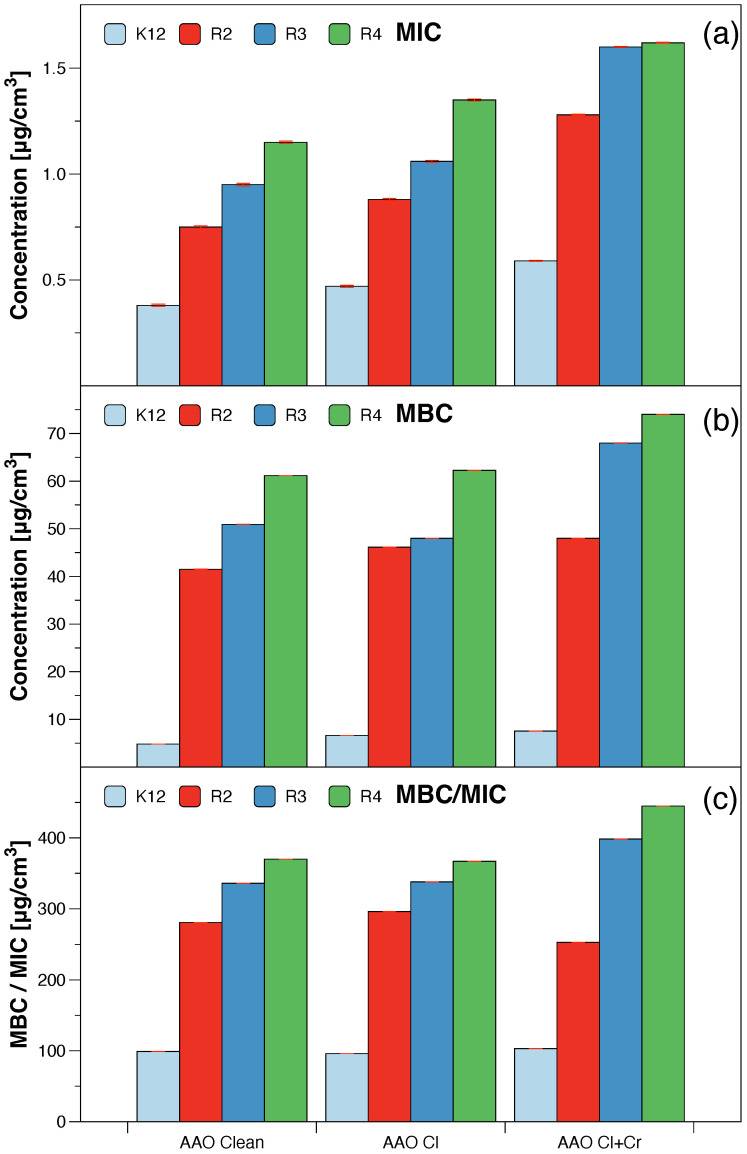
The antibacterial properties of the AAO membranes: (**a**) minimal inhibitory concentration of the AAO membranes in model bacterial strains, (**b**) minimal bactericidal concentration of the amidoximes in model bacterial strains, (**c**) MBC/MIC of the coumarin derivatives. Each experiment was performed independently in three replications (n = 3).

**Figure 4 molecules-28-00401-f004:**
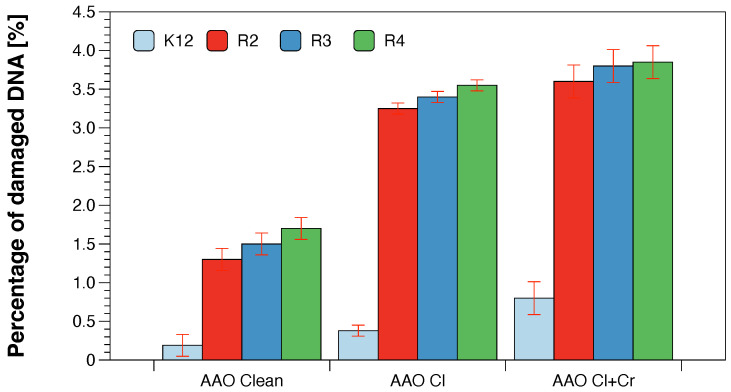
Percentage of plasmid DNA recognised by Fpg enzyme (*y*-axis) with model bacteria, K12, and R2–R4 strains (*x*-axis). Each experiment was performed independently in three replications (n = 3).

**Figure 5 molecules-28-00401-f005:**
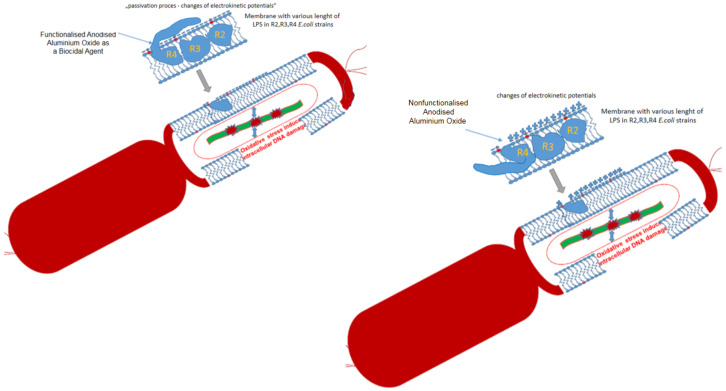
The proposed mechanism of antibacterial action of functionalised and nonfunctionalised anodised aluminum oxide in selected model *E. coli* strains.

**Figure 6 molecules-28-00401-f006:**
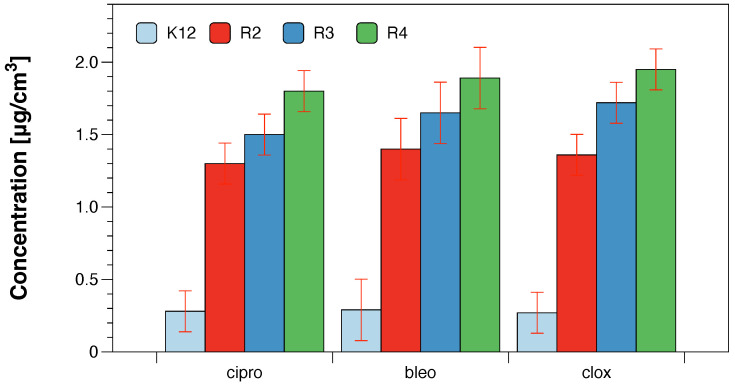
Examples of MIC with model bacterial strains K12, R2, R3, and R4 with ciprofloxacin, bleomycin, and cloxacillin. Each experiment was performed independently in three replications (n = 3).

**Figure 7 molecules-28-00401-f007:**
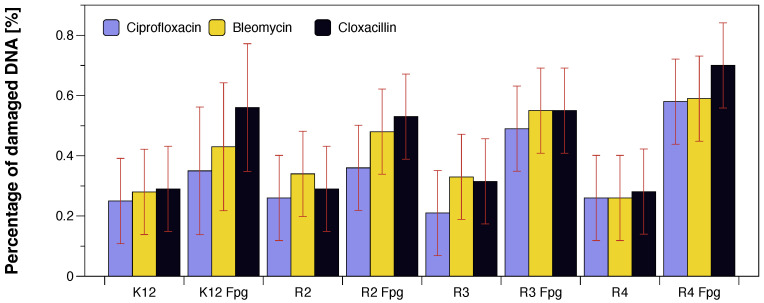
The percentage of bacterial DNA recognised by Fpg enzyme in model bacterial strains after ciprofloxacin, bleomycin, and cloxacillin treatment. The compounds were statistically significant at *p* < 0.05. Each experiment was performed independently in three replications (n = 3).

**Table 1 molecules-28-00401-t001:** The statistical analysis of the measured membranes by MIC, MBC and MBC/MIC; <0.05 *, <0.01 **, <0.001 ***.

Strain	AAO Clean	AAO + Cl	AAO Cl+Cr	Type of Test
K12	*	**	**	MIC
R2	*	**	**	MIC
R3	*	**	**	MIC
R4	*	**	**	MIC
K12	***	**	**	MBC
R2	***	**	**	MBC
R3	***	**	**	MBC
R4	***	**	**	MBC
K12	**	*	*	MBC/MIC
R2	**	*	*	MBC/MIC
R3	**	*	*	MBC/MIC
R4	**	*	*	MBC/MIC

**Table 2 molecules-28-00401-t002:** The pore size and average distance between them for the anodised membranes.

Sample	Pore Size [nm]	Distance between Pores [nm]
*AAO Clean*	34 ± 9	80 ± 19
*AAO Cl*	44 ± 12	103 ± 13
*AAO Cl+Cr*	47 ± 12	103 ± 6

**Table 3 molecules-28-00401-t003:** The chemical analysis of the *AAO Cl+Cr* membrane.

Element	C-K	O-K	Al-K	Cl-K	Cr-K	Au-M
Weight %	2.88 ± 0.16	27.12 ± 3.17	34.12 ± 0.99	0.01 ± 0.01	0.02 ± 0.02	35.85 ± 2.37
Atom %	7.10 ± 0.35	49.98 ± 3.30	37.50 ± 2.62	0.01± 0.01	0.01 ± 0.03	5.41 ± 0.57

## Data Availability

Not applicable.

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
