# Peer review of "Aluminium(III) Oxide—The Silent Killer of Bacteria"

_molecules, 2023, doi:10.3390/molecules28010401_

Round 1
Reviewer 1 Report
English language must be revised.
novelty is unclear and introduction part should be more simplified.
if possible, sem/tem images are required.
more discussion and citation should be included on previous work.
Author Response
Thank you very much for all the questions and very accurate suggestions that contributed to improving the quality and substantive value of our manuscript
Reviewer 1
- English language must be revised.
Thank you for your remark. We have thoroughly re-checked grammar, style and orthography.
- Novelty is unclear and introduction part should be more simplified.
Many researchers or even industry working with anodised aluminium(III) oxide consider the material as biologically inert. The main goal of the article is to disprove this false theory. The argument could said that the antibacterial properties derive from the biologically harmful precursors used during the process of anodisation, namely chromium and chlorine. We have also taken this into account and performed the tests on samples fabricated without them. As shown, all of the aforementioned types of anodised aluminium(III) oxide exhibit antibacterial action and are definately not biologically inert.
- If possible, sem/tem images are required.
Currently, we are not able to provide TEM images. However, SEM images of all the samples are and were already provided. Please see Figure 2.
- more discussion and citation should be included on previous work.
We have included this in our literary quotations to avoid unnecessary repetition
Reviewer 2 Report
The article presents an important proposal of different possibilities of application of nanoporous materials, demonstrating their antibacterial properties.
In materials and methods, describe the conditions used in electron microcope analyses and energy dispersive X-ray spectroscopy
In item 2.1 microorganisms and media briefly describe the methodology used In item 3.2. Antibacterial properties I suggest replacing the term flora with microbiota. In item 3.2 on antimicrobial properties, can changes be justified by differences in lipopolysaccharide chains even though the tested bacteria are from the same species? Did the isolates present the same cultivation time for carrying out the experiments?What would be the proposals for future research to try to explain the differences between the three types of membrane in terms of biocidal action?
Figure 1 was not mentioned in the text.
Author Response
Thank you very much for all the questions and very accurate suggestions that contributed to improving the quality and substantive value of our manuscript
Reviewer 2
- In materials and methods, describe the conditions used in electron microcope analyses and energy dispersive X-ray spectroscopy.
Thank you for this remark. We have indeed not incuded this information. It was added in the Materials and Methods section as suggested.
- In item 2.1 microorganisms and media briefly describe the methodology used
2.1. Microorganisms and Media
- coli K-12, R1–R4 strains were received from Prof. Jolanta Łukasiewicz at the Ludwik Hirszfeld Institute of Immunology and Experimental Therapy (Polish Academy of Sciences, Warsaw, Poland). The reference bacterial strains of E.coli (K12 ATCC 25404, R2 ATCC 39544, R3 ATCC 11775, R4 ATCC 39543 were provided from (LGC Standards U.K.) and were used according to the recommendation of ISO 11133: 2014. These strains were used to test antibacterial activity of the synthesized agents [20-30]. Bacteria were cultivated in a tryptic soy broth (TSB; Sigma-Aldrich, Saint Louis, MI, USA) liquid medium and on agar plates containing TSB medium at 25 °C. Alternatively, TSB agar plates were used. The specific growth rate (μ) according to first-order kinetics was measured using a microplate reader (Thermo, Multiskan FC, Vantaa, Finland) at 605 nm in TSB medium. Lanes 1kb-ladder, and Quick Extend DNA ladder (New England Biolabs, Ipswich, MA, USA), with MIC and MBC tests as described in detail in the previous work our manuscripts and analyzed by the Tukey test indicated by (p < 0.05): * p < 0.05, ** p < 0.1, *** p < 0.01.[20-30]. Model strains of E.coli were plotted in all 48-well plates observed; K12, R2-R4 which were treated with the analyzed compounds. From analysis of the MIC and MBC assays, color changes were observed for all compounds tested, but at different levels and at different dilutions. Bacterial strains R3 and R4 were the most susceptible to modification with these compounds due to the increasing length of their LPS (visible dilutions of 10-3 corresponding to a concentration of 0.25 µM) than strains K12 and R2 (visible dilutions of 10(-6) corresponding to a concentration of 0.06 µM). The analyzed R4 strain was the most sensitive of all strains, probably due to the longest length of the lipopolysaccharide chain in the bacterial membrane. In all analyzed cases, the MBC values were approximately 170 times higher than the MIC values .
- In item 3.2. Antibacterial properties I suggest replacing the term flora with microbiota.
Phrases have been replaced as suggested by the Reviewer
- In item 3.2 on antimicrobial properties, can changes be justified by differences in lipopolysaccharide chains even though the tested bacteria are from the same species?
Yes, because they differ in the degree of length of the LPS and the measure of its reactivity as described in previous works [..]
Did the isolates present the same cultivation time for carrying out the experiments?
Yes, the reaction conditions were identical in all analyzed strains.
- What would be the proposals for future research to try to explain the differences between the three types of membrane in terms of biocidal action?
The toxicity of the analyzed membranes with the appropriate strains of bacteria depends on their interaction with the membrane, which may affect the destruction of cell walls by changing their hydrophobicity. Changes in the permeability and integrity of the bacterial membrane may result in a specific bacterial response to biologically active compounds, e.g. antibiotics used. This may suggest that the presence of these compounds has a toxic effect on bacterial LPS, generating strong oxidative stress, which we observed in our previous studies [20-30]. The results of the presented studies are important for understanding the biological properties of the tested membranes as a function of potential new antibiotics: kanamycin, streptomycin, ciprofloxacin, bleomycin and cloxacillin commonly used in nosocomial infections with pathogenic strains causing various diseases: syphilis, sepsis or psoriasis. The use of membranes will multiply their toxic effect on Gram-negative and Gram-positive bacteria in the face of the growing drug resistance pandemic.
Based on our knowledge, we can precisely design special membranes with high selectivity of tissue functions. Currently, one of the most important research tasks is a broader understanding of the mechanisms of interaction of these compounds on various structures, both at the cellular and molecular level. Particular attention should be paid to their use in air conditioners located in office, hospital or food rooms.
There is still a need for a more fundamental understanding of the performance of already synthesized and newly designed membranes in vivo using high-throughput bacterial screening in both Gram positive and negative systems. Another practical application of the different membranes will be understanding their physiological effects under environmental conditions. In order to introduce them commercially to the market, further research is required on their biostability, half-life, preferred method of administration and potency on the analyzed bacterial, fungal or viral strains.